# Communication System Based on Magnetic Coils for Underwater Vehicles

**DOI:** 10.3390/s22218183

**Published:** 2022-10-26

**Authors:** Giovanni Canales-Gómez, Gloria León-Gónzalez, Neguib Jorge-Muñoz, José Humberto Arroyo-Núñez, Elba Dolores Antonio-Yañez, Rafael Stanley Núñez-Cruz

**Affiliations:** Control and Design Laboratory, Polytechnic University of Tulancingo, Tulancingo de Bravo 43629, Mexico

**Keywords:** wireless communication, magnetic field, underwater communication, magnetic coil, magnetic induction

## Abstract

In this work, a wireless communication system based on magnetic coils for underwater vehicles is presented. Firstly, the mathematical model of magnetic field induction using magnetic coils is discussed. Then, a description of the proposed communication system is presented, including the main components of the transmitter and receiver module. The experimental results show that due to the properties of the magnetic field, the proposed communication system can work properly in different environments such as air or water with the same efficiency. Underwater tests were carried out in different water circumstances: varying the temperature in a range from 10 °C to 35 °C, varying concentrations of clay in a range from 0% to 10%, and varying the salinity concentration in a range from 1000 ppm ( parts per million) to 35,000 ppm. It was observed that these conditions do not affect the information transfer. Finally, the advantages of using the proposed system compared to existing submarine communication systems are discussed.

## 1. Introduction

Underwater robotics has become a research area of great interest in recent years; its applications extend to different disciplines such as monitoring marine biology [1], reconstruction of the seabed [2], inspection of underwater pipelines [3], and continuous monitoring of marine species [4], among others.

One of the main challenges of underwater robotics is communication systems. The attenuation that electromagnetic signals suffer underwater makes it very difficult to use communication and global positioning systems that are commonly used on the surface [5], making it difficult to plan optimal routes in the navigation of autonomous mobile robots [6]. Electromagnetic waves have an electrical and a magnetic component, which are exposed to optical phenomena such as refraction, reflection, dispersion, and diffraction which modify the propagation pattern of the waves, causing a distortion in the received signal relative to the emitted signal.

Due to these difficulties, current submarine communication systems are very expensive, consume large amounts of energy, are relatively large, send and receive information at very low speeds, and have a limited range.

Underwater communication systems can be classified according to the technology on which they are based. The main characteristics of each technology are presented below:

### 1.1. Radio Frequency Communication

This technology is intended for communication and data transmission over long distances and is commonly used in telephones, radios, televisions, etc. These communication systems use electromagnetic waves. However, seawater has a strong conductivity (4 S/m on average) that increases especially with salinity. This is reflected in a greater attenuation (~1 dB/m), which causes a shorter range and significantly reduces the usefulness of underwater radio frequency communications [7].

An example of the application of this type of technology is low-frequency RFID (radio frequency identification) systems (120–134.2 kHz). These can be used in commercial devices for the identification and monitoring of aquatic species and has been particularly tested in salmon [8]. Another application is the use of global positioning system (GPS) and satellite communications, which have been used to follow the trajectory of marine species that travel long distances, such as dolphins, turtles, and whales. However, due to the great attenuation of electromagnetic signals, the system is only able to operate when the species is on the surface [9].

### 1.2. Optical Communication

This type of technology uses electromagnetic waves with wavelengths between 400 nm y 500 nm. Optical signals are rapidly absorbed in water because light penetrates a few hundred meters in clearer waters and less in turbid waters, suspended particles cause considerable optical scattering, and the high level of ambient light at the top of the water column interferes with communication [10].

An application of this technology is a system capable of transmitting at 5 Mbps using light-emitting diode (LED) sources for information transmission [11]. Another example is the implementation of an enhancement to the pulse position modulation (PPM) method to improve data transmission performance in a submarine optical communication system [12].

### 1.3. Acoustic Communication

Acoustic waves are mechanical vibrations that propagate in a medium. Acoustic waves suffer less attenuation relative to the communication systems described above (~1 dB/km) [13]. However, this type of communication has other limitations, such as disturbance from a higher ambient noise level, unwanted reflections that take advantage of the same conditions for easy propagation, and lower data transmission rates.

An application of this underwater communication technology is a network that coordinates autonomous underwater vehicles for mine inspection [14]. Additionally, acoustic communication has been used to exchange data between an unmanned surface vehicle and an autonomous underwater vehicle [15]. Another important application is the use of acoustic communication signals in an autonomous underwater vehicle to characterize the seafloor [16].

Table 1 shows a summary of the characteristics of the technologies used in submarine communication [17].

It has been observed that when implementing radio frequency communications, high data rates are achieved but are limited to a strong attenuation by the conductivity of seawater. On the other hand, optical communications provide much higher data rates. However, they are subject to seawater turbidity attenuation. Moreover, acoustic waves provide long transmission distances but have low data transmission rates and are subject to reflections due to multipath propagation.

Due to the limitations of the current technology used in submarine communication, the design of a communication system based on rectangular magnetic coils is proposed. This system works through magnetic induction (MI), a technology that works just as efficiently underwater as it does on land.

### 1.4. Underwater Communication with Magnetic Fields

A promising alternative for near/medium-range communication is based on magnetic fields [18]. By using a transmitter and a receiver coil, the information signal sent is reproducible on the receiver side. When compared to acoustic communication, the advantages of MI include negligible propagation loss and less susceptibility to surroundings. This technique is insensitive to water turbidity (including wave zones, tides, river flow), depth of water, reflections, and interference from sound and light. Furthermore, rapidly decaying magnetic fields are more environmentally friendly for marine mammals than sound waves [19].

This technology is already being developed and used in various projects. For instance, it has been applied as a complement to a cooperative multiple-input multiple-output (MIMO) communication mechanism that uses a hybrid acoustic and magnetic induction technique [20]. Another example of its use is a new approach for hybrid submarine communication nodes based on magnetic induction and visible light which presents an improvement in the position estimation of autonomous underwater vehicles [21].

The content of the article is presented as follows: in Section 2, the basic principles of the mathematical modeling of magnetic coils are presented; Section 3 presents the description of the elements of the proposed communication system, including the transmitter and receiver system; Section 4 presents the main characteristics of the test platform used to evaluate the submarine communication system; Section 5 presents the experimental results of the communication system and finally, the conclusions and references are presented.

## 2. Magnetic Field Induction Using Coils

Any conductor through which an electric current circulates generates a magnetic flux; the relationship between the magnetic flux and the current that circulates is an electrical property known as inductance, which is measured in Henry (H). As the current (amount of moving charge) is increased, the magnetic field increases proportionally. If the observation point is moved away from the conductor, the field decreases inversely proportional to the distance [22].

The analysis of a rectangular magnetic coil that uses thin conductors can start from the physical phenomenon of the generation of a magnetic field $\left(\vec{B}\right)$ due to the flow of electric current through the conductor. Maxwell’s equations indicate that the divergence $\vec{\nabla }\cdot \vec{B}=0$, which indicates that $\vec{B}$ has the solenoidal property (no divergence).

$\vec{B}$ can be presented by using an auxiliary vector function $\vec{A}$, through the rotational relationship, as in Equation (Equation 1).
(1)$$\vec{B}=\vec{\nabla }\times \vec{A}$$

Function $\vec{A}$ is called a magnetic vector potential field. This field is related to the sources of stable current density responsible for the magnetic field $\vec{B}$. For the case of a thin linear conductor, $\vec{A}$ is defined by Equation (Equation 2).
(2)$$\vec{A}=\frac{\mu_{0}\vec{I}}{4\pi }\int_{l}\frac{dl^{\prime }}{r}$$
where $\vec{I}$ is the current flowing through the linear conductor and *r* is the distance from the conductor to the analysis point. For a rectangular coil of thin conductor of *n* turns, the magnetic field is calculated by adding the contribution generated by each segment of the coil. In Figure 1, the magnetic coil is shown in the plane XY, and the analysis point of the magnetic field P(X,Y,Z).

The calculation of the field $\vec{B}$ must be done using Equation (Equation 2) that implies the integral over the geometry defined by the emitting coil, each of the 4 segments of linear conductor of each of the *n* turns considered in Figure 1.

As an example, the calculation of the field ${\vec{A}}_{1}$ is shown, which represents the first segment of the coil on the plane XY that goes from the coordinates (a,−b,0) to the point (−a,−b,0) [23]:(3)$${\vec{A}}_{1}=\frac{\mu_{0}\vec{I}}{4\pi }\int_{a}^{-a}d\frac{dx^{\prime }}{\sqrt{{\left(x-x^{\prime }\right)}^{2}+{\left(y+b\right)}^{2}+z^{2}}}$$

The vector magnetic potential field $\vec{A}$ is calculated as the sum of the fields of all the segments ${\vec{A}}_{k}$. Later, the field $\vec{A}$ can be evaluated with Equation (Equation 2). Finally, to find the magnetic field, $\vec{B}$ Equation (Equation 1) is used.

Figure 2 shows the simulation results of the magnetic field generated by a coil using the following parameters: *a* = 0.12 m, *b* = 0.10 m, *I* = 70 mA, *n* = 17, and height *z* = 0.05 m.

The magnetic field generated by the current that circulates in a conductor (transmitter coil) can be induced to another conductor (receiver coil) that will in turn generate electric current.

The induced voltage or electromotive force (EMF) can be obtained by taking the derivative with respect to time of the integral of the magnetic field $\vec{B}$ along the geometry defined by the receiver coil, according to Equation (Equation 4).
(4)$$EMF=-\frac{d}{dt}\int_{s}^{}\vec{B}\left(I\left(wt\right)\right)ds$$
where *s* is the trajectory of the receiver coil including all the linear segments. The electrical current is defined in terms of the frequency *w* of the periodic signal used.

Figure 3 shows the results of the voltage induction simulation are shown using the following parameters: *n* = 40, *a* = 0.12 m, and *b* = 0.10 m for the receiver coil.

In Figure 4, the comparison between experimental and simulated values of magnetic field is shown. To obtain the experimental measurements a magnetic field meter (model ELT-400 manufacturer Narda imported from Germany) was used, making a sweep in the *X* axis in the center of the transmitter coil, taking samples every 2.5 cm of separation. An error of less than 5% was obtained.

Due to the propagation restrictions of the magnetic field, it was necessary for both coils to be close enough to each other to achieve transmission. There are applications in which this property results in an advantage, such as when transmitting geolocation data in a precise area [24], when transmitting bank information between devices, the transfer of information between mobile devices, access control [25], biomedical applications (e.g., health care and disease monitoring) [26], and classification of vehicles or containers [27].

## 3. Description of the Communication System

As previously shown in Equation (Equation 4), the frequency of the current signal is one of the factors that determines the magnitude of the magnetic field generated by the coil. For this reason, modulated signals are used in coil data transmission applications.

Modulation is the name given to the operation where certain characteristics of a wave called “carriers” are modified according to another wave called modulator, which contains information, so that the latter can be transmitted. The wave obtained is called the modulated signal.

For this application, the modulation algorithm known as amplitude shift keying was used. It is a digital modulation technique consisting of switching on and off a carrier sinusoidal signal by means of a modulating signal. Its main advantage is that the processes of modulation and demodulation are relatively cheap.

Taking into account the aforementioned considerations, a communication system based on magnetic field induction is proposed, which uses the serial protocol to communicate two devices submerged under water and the data signal is transmitted in a modulated manner. In order to prove the concept, unidirectional communication is presented. Below is a description of the transmitter and receiver systems.

### 3.1. Transmitter System

Figure 5 shows the most important stages of the transmitter system.

#### 3.1.1. USB ↔ Serial Converter

A commercial device used to emulate a serial port in the sending system’s computer. The data from the serial converter are used as the modulating signal.

#### 3.1.2. Carrier Signal Generation

A Schmitt trigger circuit was used to build a sinusoidal signal that will serve as the carrier signal. This configuration offers variable frequency thanks to the built-in potentiometer trimpot trimmer 3296 W of 100 kΩ and logic gate 74LS132. This signal has an amplitude of 2.5 V and a frequency of 600 kHz. The carrier signal can be seen in Figure 6.

#### 3.1.3. Modulation

To implement the modulation, the integrated circuit CD-4066BE was used, which is a quadruple bilateral switch with independent controls using a single channel controlled by the modulating signal.

Figure 7 shows a comparison of the digital information signal (pink color) and the corresponding modulated signal (yellow color).

#### 3.1.4. Signal Conditioning

The last part of the transmitter system corresponds to the conditioning of the modulated signal. In this part, the voltage and current are amplified to adjust the desired transmission range. The amplification process is carried out by connecting two NPN-type transistors 2N2222A in cascade, obtaining a common-emitter two-stage amplifier to obtain current gain.

### 3.2. Receiver System

Figure 8 shows the most important stages of the receiver system.

#### 3.2.1. Signal Conditioning

The voltage induced in the receiving coil is in the order of millivolts, so it is necessary to carry out an amplification of voltage and current. In this stage, the receiving magnetic coil is connected to ground and to the input of the voltage amplifier circuit and the output is directly connected to the base of a 2N2222 transistor directly configured to have current amplification.

#### 3.2.2. Demodulation

The signal obtained in the receiver coil must be demodulated to obtain logic levels that can be correctly interpreted in a microcontroller. This stage was carried out using an envelope detector, avoiding frequency and phase problems that appear in synchronous detection. In this case, the signal that arrives at the receiver coil is passed through a non-linear device (diode 1N4001) and a low pass filter, the diode rejects the negative part of the amplitude shift keying signal, and the low-pass filter cannot follow the variations of the signal, so it only keeps the envelope of the signal that the diode allows to pass.

Figure 9 shows the comparison between the emitted signal prior to the modulation stage (yellow color) and the received signal after the demodulation stage (pink color). In this figure, it can be seen that communication between the two systems is established.

Communication was established adequately through the air, and the transmitted information was verified using laboratory instruments. To evaluate the application of the proposed system for underwater vehicles communication, a submersible module was designed and attached to an underwater robot as shown in following section.

## 4. Experimental Tests

To verify the proposed communication system, the printed circuit boards of the transmitter and receiver circuit described in the previous section were built as shown in Figure 10.

To evaluate the underwater communication system, a submersible transmitter module was built, which uses a 14.8 V battery, a regulator, n embedded Raspberry Pi computer, a USB↔serial converter, a transmitter circuit, and a transmitter coil.

The receiver system was placed in the prototype of an underwater vehicle based on the BlUEROV2 robot, which has been used in different applications such as evaluation of control laws [28] and marine research [29]. The mechanical structure of the vehicle is made of polyethylene and acrylic tubes where the electronic components are located.

The underwater vehicle has an embedded Raspberry pi computer to implement the algorithms for control, navigation and signal processing. The system works using the Ubuntu operating system under the Robot Operating System (ROS) programming environment, an open-source code environment developed especially for robots. Under this architecture, programs were designed to access data from the pressure sensor, inertial center, transmission of control signals for the thrusters, teleoperation signals, etc.

Regarding the interface with the proposed communication system, a Python program was developed that connects to the universal asynchronous receiver-transmitter (UART) serial port of the USB↔serial converter. In this way, it is possible to access the data read by the receiver magnetic coil.

A program was also designed that stores the received data in a text file, which allows for its subsequent analysis.

The receiver coil was mounted on the front of the mechanical structure of the underwater robot, the receiver circuit was placed inside the capsule in which the robot’s computer is located and the connection is made using a hermetic connector, the coil is connected to a separate battery that is used by the underwater vehicle’s control system.

Figure 11 shows the description of the experimental tests which are defined as follows:1.Start the transmitter system, which must be programmed to periodically send a message containing a password and a string of test characters.2.Submerge the emitter module to a depth of at least 1.5 m.3.Activate the underwater robot and launch the teleoperation and data reading programs. The latter must be connected to the port where the receiver system is located.4.Submerge, through teleoperation, the underwater vehicle in such a way that the coils are at a distance of approximately 20 cm, horizontally.5.Activate an information storage program which should save the string of test characters received from the transmitter module in a file for further analysis.

The use of a password is recommended during the transmission tests to avoid storing erroneous data produced by noise. It is recommended to submerge the emitter module at least 1.5 m to ensure that the magnetic field lines generated propagate only in water. Finally, it is also recommended to place the receiver coil at a maximum distance of 20 cm from the transmitter coil to ensure that it is placed within the coverage area of the transmitter system (this distance may change depending on the characteristics of the magnetic coils used for transmission).

## 5. Results

To evaluate the system’s performance, data transmission tests were performed following the description of the previous section. Two square coils were built according to the specifications in Table 2.

During the test, the emitted signal consumed 75 mA with a voltage of 2.5 V at a frequency of 600 kHz. The magnetic field generated has a maximum magnitude of 1.2 uT, so a transmission range of up to 45 cm is observed. The induced voltage at 20 cm is 260 mV.

The test message is a text string with 100 characters. This message was sent consecutively 1000 times for each position sampled at a transmission speed of 9600 bps. In Table 3, the number of times the test message was transmitted incorrectly is displayed. Each box corresponds to different locations of the receiver coil, using the center of the transmitter coil as a reference.

When the underwater vehicle is positioned in the center of the transmitter coil, the data reception does not contain errors; these appear when the submarine is about to leave the coverage area and there is no communication when both coils (transmitter and receiver) stop sharing parallel planes.

With the data obtained in Table 3, the data transmission area was plotted and shown in Figure 12a. It can be observed that for the information transfer to be carried out successfully, it is sufficient for a part of the receiver coil to enter the indicated space.

Table 4 shows the number of messages transmitted incorrectly. When the submarine is placed within the coverage area but with $45^{\circ }$ of rotation, the data are received satisfactorily without affecting the transmission of information.

In Figure 12b, the data transmission area for the second test is plotted, as can be seen the transmission is not affected by changes in orientation around the axis of the coils.

Additional tests were carried out while varying the temperature of the water; modifying the level of salinity concentration; adding impurities such as clay, cardboard, plastic, and glass; and without these variables modifying the behavior of the system.

The temperature tests were carried out in a range from 10 °C to 35 °C. The measurements were taken in real time using a MS5837-30BA sensor, which has a precision of ±1 °C. By measuring the amount of salt in a given amount of water, we can examine the concentration of salt in it. Concentration is the amount (by weight) of salt in the water, which can be expressed in parts per million (ppm). For the salinity tests, transmission tests were made by mixing salt directly into the water in a range from 1000 ppm to 35,000 ppm. For the tests with different concentrations of clay, this element was mixed directly, from 0 g to 1 kg.

When carrying out data transmission tests, it was observed that it is necessary to place the receiver magnetic coil at least 15 cm away from the closest propeller to avoid interference due to the magnetic field that it generates when it is in operation.

Figure 13 shows a test run according to the specifications of the previous section. Due to the fact that the field lines travel through air or water and the magnetic permeability values of these elements differ, the information does not suffer distortion due to the change of medium. For this reason, this communication system can send and receive information underwater with the same efficiency as it does in the air.

## 6. Conclusions

In the present work, a communication system based on magnetic coils of rectangular geometry was opened and implemented, and the electronic circuits necessary to transmit data of a serial communication protocol through water were designed.

Several transmission tests were carried out while modifying the water temperature, varying the salinity concentration, and adding impurities such as cardboard, plastic, glass, and clay in different concentrations, and it was concluded that the aquatic environment does not present additional difficulties for data transmission.

The proposed system has advantages over acoustic technology since it does not lose information due to the multipath effect, it is easier to identify the emitter, it is cheaper to implement, and it reaches higher data transmission speeds.

It has advantages over systems based on radio frequency since it is not affected by the salinity level of the water, it uses less energy, and the dimensions of the transmitter and receiver coils are smaller, achieving greater versatility.

The proposed system using magnetic fields is better than optical systems, because it is not affected by the turbidity of the water and can operate in water with different concentrations of clay.

The proposed system has some advantages with respect to similar works (presented in Section 1.4): The communication system presented in [19] uses up to 7 A to power the emitter system, while our system is capable of operating with only 75 mA, achieving greater autonomy. It transmits information at higher speeds in the range of 9.6 kb/s to 115.2 kb/s compared to 21.64 kb/s or less reported in [20,21]. Building the transmitter and receiver module is cheaper because it is made up of simpler components.

Therefore, the proposed system can be consider as a viable option to be used in underwater vehicles for a wide range of applications in data transmission such as transferring information between a vehicle and an underwater station, accurate spacial localization, tracking trajectories, exploration and mapping of the seabed, induction of voltage to another vehicle to recharge its batteries, etc.

In future work, we will try to use the data obtained with the underwater vehicle to carry out a mapping mission automatically, that is, use the transmitter system to send position and orientation data to the underwater vehicle, modify the programming so that these data are converted in reference signals, and make the robot know its position and manage to follow a programmed route. In order for the robot to move without a remote operator, it is essential to apply an automatic control law to control orientation and depth simultaneously and use artificial vision to be able to move faster towards the coverage area of the transmitter magnetic coil and avoid obstacles that could potentially damage or obstruct the path of the underwater vehicle.

## Figures and Tables

**Figure 1 sensors-22-08183-f001:**
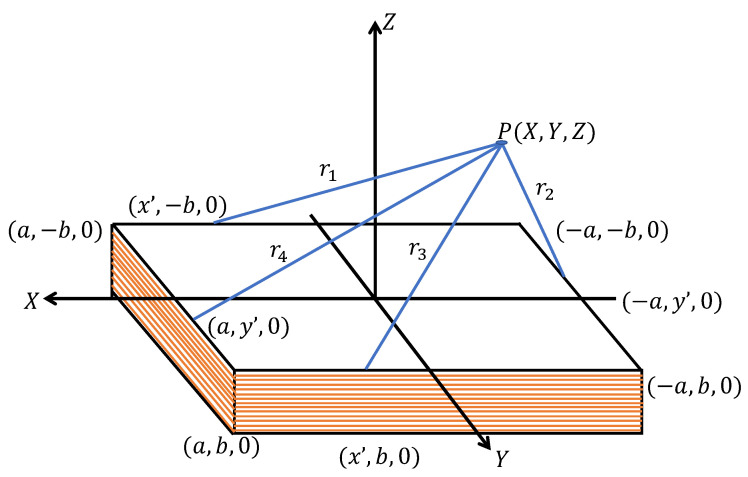
Square magnetic coil and point of interest to analyze.

**Figure 2 sensors-22-08183-f002:**
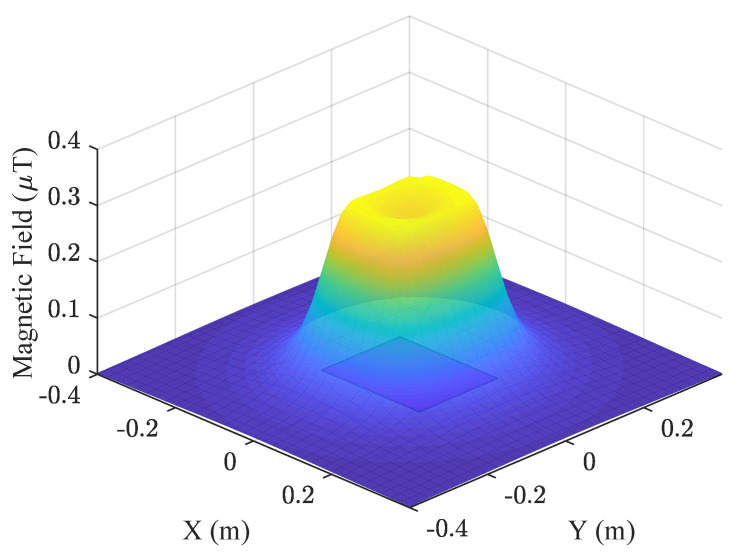
|B| in an area of 0.4 × 0.4 m at a height of *z* = 0.05 m.

**Figure 3 sensors-22-08183-f003:**
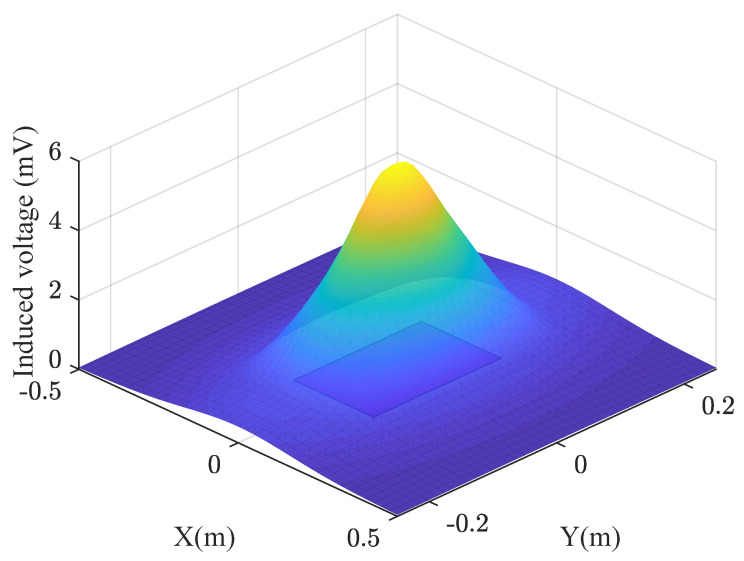
Induced voltage at a height of 0.2 m.

**Figure 4 sensors-22-08183-f004:**
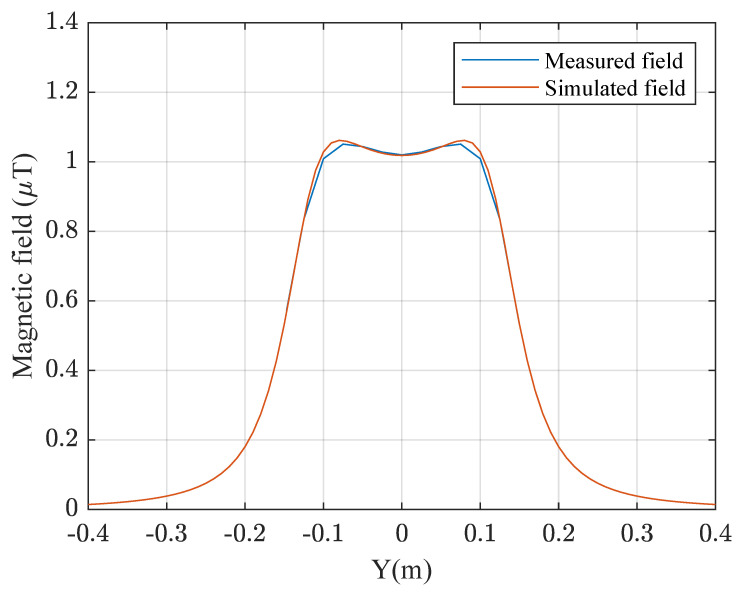
Magnetic field measured with ELT-400 and simulated.

**Figure 5 sensors-22-08183-f005:**
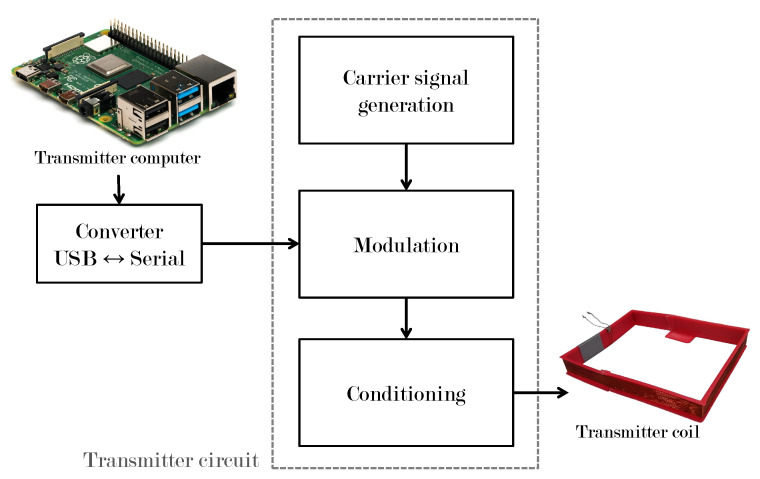
Transmitter system.

**Figure 6 sensors-22-08183-f006:**
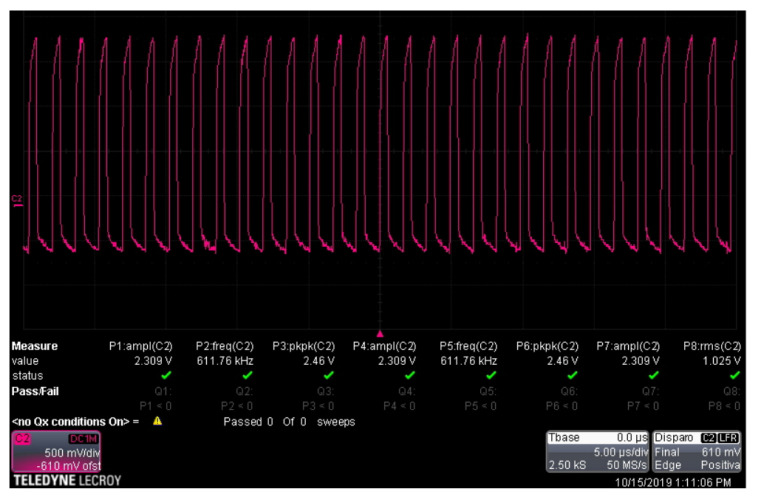
Carrier signal.

**Figure 7 sensors-22-08183-f007:**
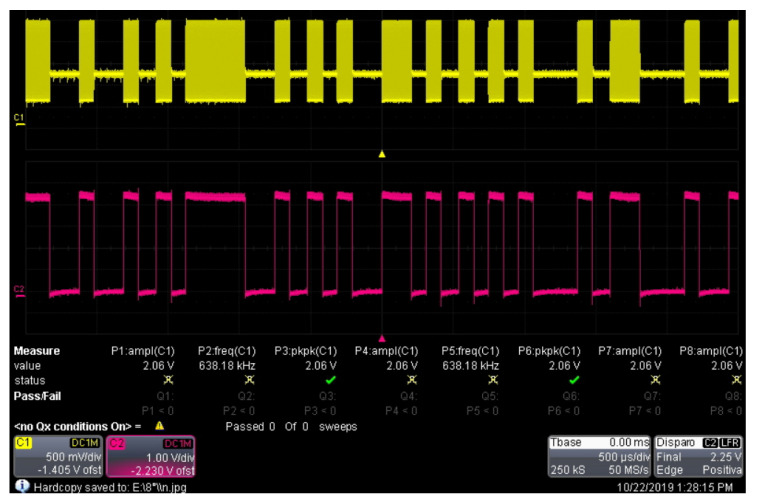
ASK modulation tests.

**Figure 8 sensors-22-08183-f008:**
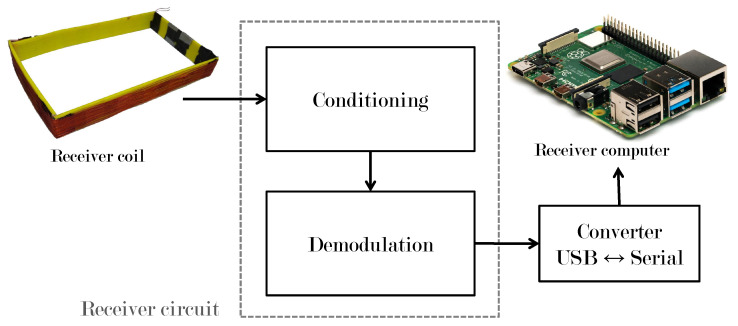
Receiver coil electrical circuit.

**Figure 9 sensors-22-08183-f009:**
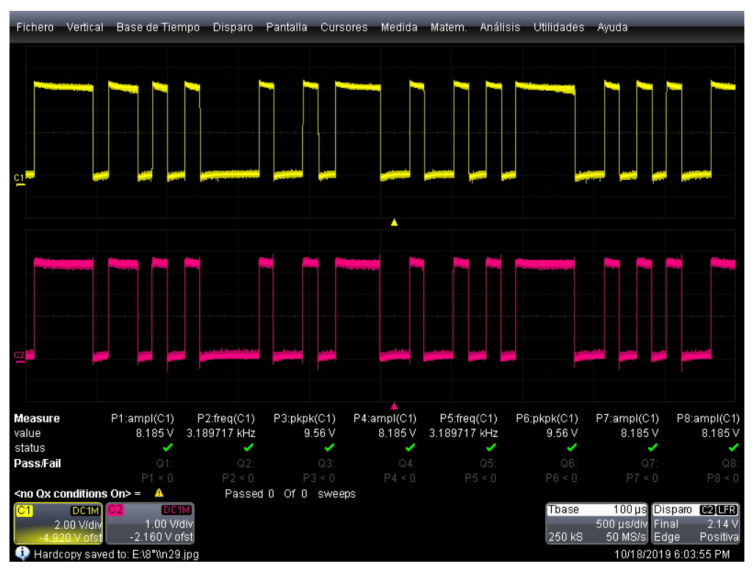
Comparison of emitted and received signals.

**Figure 10 sensors-22-08183-f010:**
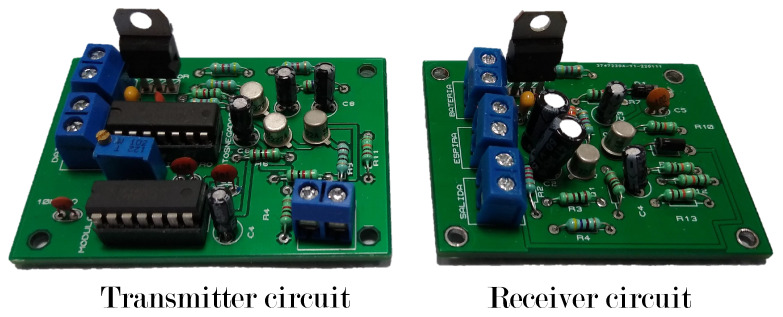
Construction of the transmitter and receiver circuits.

**Figure 11 sensors-22-08183-f011:**
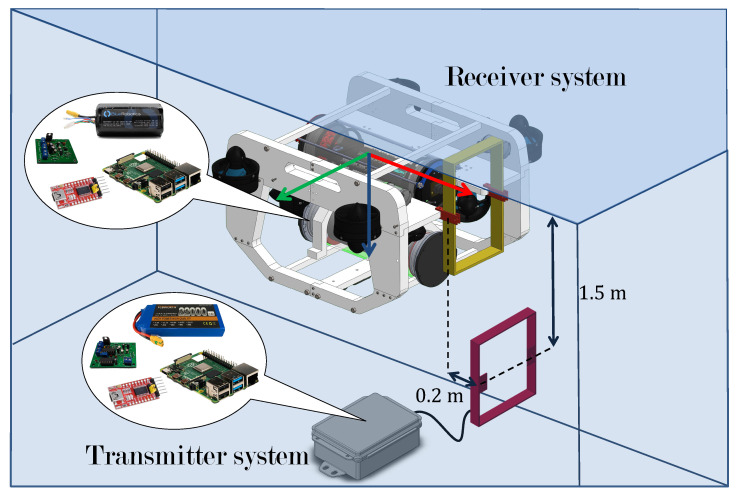
Experimental test system.

**Figure 12 sensors-22-08183-f012:**
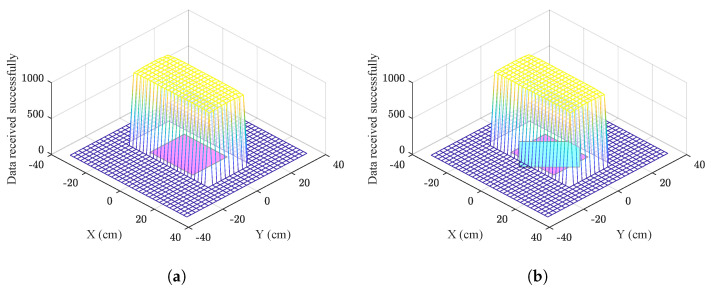
Data successfully received underwater. (**a**) Coils aligned. (**b**) Coils rotated 45°.

**Figure 13 sensors-22-08183-f013:**
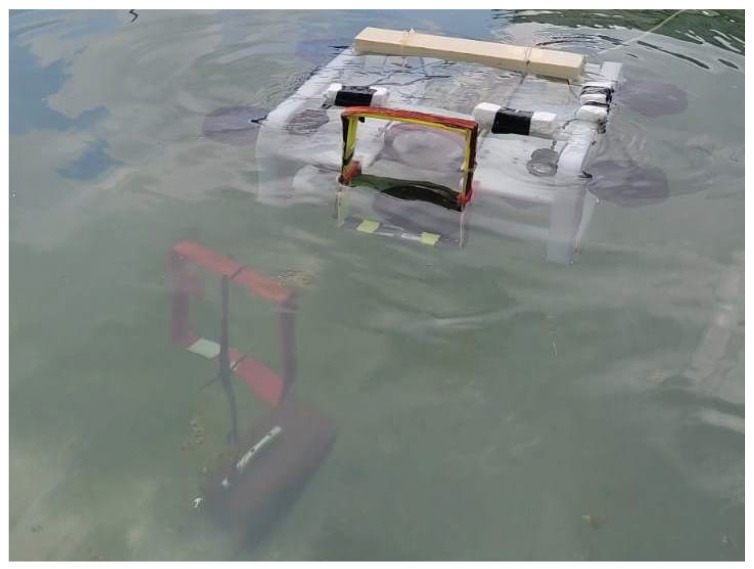
Data transmission in water with submarine.

**Table 1 sensors-22-08183-t001:** Comparison of underwater communication systems.

	Acoustics	Radio Frequency	Optics
Speed of sound	~1500 m/s	~300,000 km/s	~300,000 km/s
Data speed	<100 Kbps	<10 Mbps	<1 Gbps
Power losses	>0.1 dB/m/Hz	~28 dB/km/100 MHz	turbidity
Bandwidth	kHz	MHz	~10–150 Hz
Frequency band	kHz	MHz	~$10^{14}$–$10^{15}$ Hz
Limitations	Bandwidth	power	ambient
Effective range	km	~1–100 m	~1–100 m

**Table 2 sensors-22-08183-t002:** Characteristics of the coils.

Coil	Transmitter	Receiver
Length *a*	0.12 m	0.12 m
Length *b*	0.10 m	0.10 m
Height	0.03 m	0.03 m
Number of loops	20	40
Conductor diameter	1.024 mm	0.4049 mm

**Table 3 sensors-22-08183-t003:** Incorrectly transmitted messages per position.

Y/X	−24	−22	−20	−18	−16	−14	−12	−10	−8	0	8	10	12	14	16	18	20	22	24
−10	1000	18	5	4	3	2	2	2	2	2	2	2	2	2	3	4	5	18	1000
−7.5	1000	16	1	0	0	0	0	0	0	0	0	0	0	0	0	0	1	15	1000
−5	1000	7	0	0	0	0	0	0	0	0	0	0	0	0	0	0	0	8	1000
−2.5	1000	5	0	0	0	0	0	0	0	0	0	0	0	0	0	0	0	6	1000
0	1000	5	0	0	0	0	0	0	0	0	0	0	0	0	0	0	0	5	1000
2.5	1000	5	0	0	0	0	0	0	0	0	0	0	0	0	0	0	0	5	1000
5	1000	6	0	0	0	0	0	0	0	0	0	0	0	0	0	0	0	7	1000
7.5	1000	18	2	1	0	0	0	0	0	0	0	0	0	0	0	0	1	17	1000
10	1000	20	16	7	5	3	2	2	2	2	2	2	3	3	4	5	17	20	1000

**Table 4 sensors-22-08183-t004:** Incorrectly transmitted messages per position at 45°.

Y/X	−24	−22	−20	−18	−16	−14	−12	−10	−8	0	8	10	12	14	16	18	20	22	24
−10	1000	17	5	4	3	2	2	2	2	2	2	2	2	2	3	4	5	18	1000
−7.5	1000	16	1	0	0	0	0	0	0	0	0	0	0	0	0	0	1	15	1000
−5	1000	7	0	0	0	0	0	0	0	0	0	0	0	0	0	0	0	8	1000
−2.5	1000	5	0	0	0	0	0	0	0	0	0	0	0	0	0	0	0	6	1000
0	1000	5	0	0	0	0	0	0	0	0	0	0	0	0	0	0	0	5	1000
2.5	1000	5	0	0	0	0	0	0	0	0	0	0	0	0	0	0	0	5	1000
5	1000	6	0	0	0	0	0	0	0	0	0	0	0	0	0	0	0	7	1000
7.5	1000	18	2	1	0	0	0	0	0	0	0	0	0	0	0	0	1	17	1000
10	1000	18	16	7	3	3	2	2	2	2	2	2	3	3	4	5	17	19	1000

## Data Availability

The results of the system design and the basic code of this work are available from the corresponding author on reasonable request.

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
