# Peer review of "Communication System Based on Magnetic Coils for Underwater Vehicles"

_sensors, 2022, doi:10.3390/s22218183_

Round 1
Reviewer 1 Report
This paper presents a communication system based on magnetic coils for underwater. vehicles. The topic of this research and output might be useful for the Sensors readers. The conducted work is worth to publish, however the text needs to be corrected.
1. The abstract should describe the content of the article as much as possible. Try to supplement it, e.g. one or two sentences, with the main results of the research carried out.
2. Throughout the article, the units of quantities are written in italics. Units are not written in italics. Only quantities are written in italics. Please change all quantities to normal font and add a space between the value and the quantity. E.g. 120 – 134.2 kHz, a = 0.12 m, b = 0.10 m, etc.
3. In subsection 1.4 you have the abbreviation MIMO. Please write its meaning in brackets.
4. On Figure 1, it would be appropriate to write individual quantities in italics. For the Z axis, you have an arrow at the end that indicates the direction. Arrows are missing for the X and Y axes, please add them.
5. The article deserves to be expanded with another chapter called Discussion. In this chapter, it would be appropriate to try to compare your research results with the research results of other authors dealing with similar issues, if these publications are available.
Reviewer 2 Report
1. After equations 2 and 4, the equation should be described with "where...", and there is no need to start a new paragraph. Please check the full text for correction.
2. There are five descriptions in Figure 11, and a space should be reserved before the number. At the same time, please indicate the basis of these experimental settings.
3. On page 12, the author conducted additional experiments on different conditions, such as water temperature and salt concentration. Can you describe the experimental data and process in detail?
4. When discuss the agent behavior of robot system, ‘A review on representative swarm intelligence algorithms for solving optimization problems: Applications and trends’ would be valuable for the description in introduction.
5. In the conclusion part, the description lacks refinement and logic, and it is suggested to further summarize the main contributions of this paper.
Reviewer 3 Report
The contributions is clear but the presentation of the work I'd poor.
The abstract require to rewrite focusing on the contribution and the work impact.
Authors should justify all the decisions that made in the article.
Reviewer 4 Report
In this manuscript, the authors presented a wireless communication system based on magnetic coils for underwater vehicles. The proposed communication system is introduced including the main components of the transmitter and receiver module. Experimental results exhibited that the proposed communication system can work properly in different environments. It is considered that the proposed system is a viable option to be used in underwater vehicles
The literature is well-described, making readers understand a brief studying history of this field. Four communication methods are introduced, i.e., radio frequency, optical, acoustic, and magnetic field communications.
The pain point is given for readers to know where is the main problem. "…current submarine communication systems are very expensive, consume large amounts of energy, are relatively large, send and receive information at very low speeds and have a limited range."
The theoretical model of magnetic field induction using magnetic coils is discussed.
In summary, the manuscript can be published after major revision.
Some suggestions are listed below.
-----------------------------------------------
-----------------------------------------------
[[Some suggestions]]
The suggestions are structured as shown below.
[Suggested point][Position]
Descriptions.
1. [Sentence recommendation][Abstract]
The first sentence “This article presents a wireless communication system based on magnetic coils for underwater vehicles.” is suggested to be modified as “In this work, a wireless communication system based on magnetic coils for underwater vehicles is presented.”
2. [Abbreviation][Page 2]
The abbreviations (GPS, LED) should be stated with the full name before using them.
3. [Abbreviation][Page 3]
The abbreviation (MIMO) should be stated with the full name before using it. Although there is a list of abbreviations in the last.
4. [Typo][Figure 5]
Please check the word “serie.”
5. [Typo][Figure 8]
Please check the word “serie.”
6. [Abbreviation][Page 9]
The abbreviations (ROS, UART) should be stated with the full name before using them.
7. [Typo][Page 9]
Please check the word “Rasberry.” It should be “Raspberry.”
8. [Distortion of the figure][Page 15]
The portrait of Giovanni Canales exhibits large distortion. Please modify it.
9. [He or She ?][Page 15]
Please check the biography of Gloria León.
“Gloria León is a student of the Doctorate program in Optomechatronics at the Polytechnic University of Tulancingo, [she] is currently developing a thesis project on underwater manipulation robotics. [He] graduated as an Electromechanical Engineer from the Technological Institute of the Sierra Norte de Puebla in 2016 and obtained a master's degree in automation and control from the Polytechnic University of Tulancingo in 2020.”
Round 2
Reviewer 4 Report
In the revised manuscript, much missing information has been added to clear the descriptions. Figures also have been modified to match the statements. Some additional descriptions have been added to enhance the major statements.
In summary, the revised manuscript fits the journal’s criteria for publication.